# Data-Independent Phase-Only Beamforming of FDA-MIMO Radar for Swarm Interference Suppression

Geng Chen [1], Chunyang Wang [1], Jian Gong [1,*], Ming Tan [2] and Yibin Liu [1]

1    School of Air Defense and Missile Defense, Air Force Engineering University, Xi'an 710051, China
2    College of Information and Communication, National University of Defense Technology,
Wuhan 430010, China
*    Correspondence: drgong@aliyun.com

**Abstract:** This paper proposes two data-independent phase-only beamforming algorithms for frequency diverse array multiple-input multiple-output radar against swarm interference. The proposed strategy can form a deep null at the interference area to achieve swarm interference suppression by tuning the phase of the weight vector, which can effectively reduce the hardware cost of the receiver. Specifically, the first algorithm imposes constant modulus constraint and sidelobe level constraint, and the phase-only weight vector is solved. The second algorithm performs a constant modulus decomposition of the weight vector to obtain two phase-only weight vectors, and uses two parallel phase shifters to synthesize one beamforming weight. Both methods can obtain the phase-only weight to realize suppression for swarm interference. Simulation results demonstrate that our strategy shows superiority in beam shape, output signal-to-interference-noise ratio, and phase shifter quantization performance, and has the potential for use in many applications, such as radar countermeasures and electronic defense.

**Keywords:** frequency diverse array multiple-input multiple-output (FDA-MIMO); beam pattern synthesis; swarm interference suppression; data-independent; phase-only control

## 1. Introduction

Beamforming is an essential technique in array signal processing and has been used to achieve interference suppression by forming a deep null in interference directions [1–6]. Currently, due to its powerful ability in signal processing, the technique has exhibited enormous potential in communication, radar, sonar and other fields. Unfortunately, the rapid development of unmanned aerial vehicles has made swarm interference a serious threat to airport aviation security [7,8]. It is important to suppress swarm interference to improve radar detection performance in the complex electromagnetic environment.

Over the past decades, we have witnessed explosive studies on numerous beamforming methods for phased array radar interference suppression, such as the classic linearly constrained minimum variance (LCMV) beamformer [9], minimum variance distortionless response (MVDR) beamformer, sampling matrix inversion (SMI) beamformer [10] and interference plus noise covariance matrix (INCM) reconstruction beamformer [11,12]. Recently, numerous studies have been conducted on data-independent beamforming methods [13–19]. Such methods do not require the acquisition of an accurate INCM matrix, and the performance of the beamformer is not affected by the sampling covariance matrix (SCM). Nevertheless, swarm interference is divided into mainlobe swarm interference and sidelobe swarm interference. The above phased array beamforming methods can only form a deep null at a specific angle and cannot suppress the mainlobe swarm interference by forming a null in a specific area.

Recently, due to the advantages of two-dimensional (2D) degrees of freedom (DOFs) in range and angle, frequency diverse array multiple-input multiple-output (FDA-MIMO)

radar has been widely investigated [20–26]. Specifically, it can form a range-angle-dependent beampattern by adding the frequency offset between the elements of the transmitter. Thereby, the swarm interference can be suppressed by controlling the beampattern to form a two-dimensional deep null in the interference area [27].

Using FDA-MIMO radar, Ref. [28] proposed a preset broadened nulling beamformer (PBN-BF) by placing artificial interferences with appropriate powers around the nulls of the equivalent transmit beampattern. The enlarged widening of the nulls guarantees the effective suppression of mainlobe deceptive interference. However, a band-shaped null was generated based on the PBN-BF algorithm at the interference transmit spatial frequency, which made it impossible to form a 2D regional null. In Ref. [29], a data-independent beamforming method was proposed, which could form a regional null by assigning artificial interferences with prescribed powers in the joint transmit–receive spatial frequency domain. It could realize point-by-point successive null broadening control (SNBC) and multi-point concurrent null broadening control (CNBC). Based on the above strategies, these data-independent methods provide a way to suppress interference by adding artificial virtual interference to the interference area. Nevertheless, the swarm interference distributed in an area, excessively adding virtual interferences, brings beam distortion in the non-controlled area when forming a two-dimensional null. In addition, these methods ignore the consideration of output signal-to-interference-noise ratio (SINR).

Furthermore, the phase-only weight vector cannot be obtained by the aforementioned FDA-MIMO radar beamforming methods, and an amplitude adjustment unit with a high dynamic range is required behind each matched filter. Therefore, the hardware architecture is complicated, and high costs are involved when implementing these methods [30,31]. In the past, a few approaches to phase-only beamforming for phased array have been reported [32–34], such as using neural networks [32], numerical optimization techniques [33] and geometric approaches [34]. However, to date, the method to suppress swarm interference using a low-cost FDA-MIMO radar system has been rarely reported, which requires the weight vector in the receiver to be constant in amplitude but different in phase.

In order to fill the above gap, we propose two data-independent phase-only beamforming methods for FDA-MIMO radar against swarm interference using the powerful convex optimization theory. Specifically, we consider two different technical approaches to obtain the phase-only weight vector. The first method imposes constant modulus constraint and sidelobe level constraint to solve the phase-only weight vector with the maximum output SINR as the objective function. The second decomposes the weight vector into constant modulus complex numbers from a different perspective and uses the dual phase receiver to synthesize the one beamforming weight. Moreover, the constant modulus decomposition method can convert other controlling complex weight beamforming methods to phase-only beamforming. Simulation experiments show that both proposed methods can form a better beam shape in the interference area and obtain higher output SINR than the conventional beamforming methods. We summarize the main contributions of this study as follows:

(1) We propose two data-independent phase-only beamforming methods for FDA-MIMO to suppress the swarm interference.
(2) Our proposed algorithms can achieve interference suppression by only tuning the phase of the weight vector, which efficiently reduces the hardware cost of the FDA-MIMO radar.
(3) Our proposed dual-phase shifter receiver for FDA-MIMO can convert other complex weight vector beamforming methods to phase-only beamforming.
(4) The proposed method can obtain excellent output SINR with a small number of snapshots.

This paper is organized as follows. In Section 2, the signal model is established, and the related parameters are defined. Section 3 recalls the adaptive beamforming methods. In Section 4, two data-independent phase-only beamforming methods are introduced. In Section 5, numerical simulations are performed to demonstrate the performance of the proposed method, and Section 6 concludes the paper.

Notations: In this paper, $(\cdot)^*$, $(\cdot)^T$ and $(\cdot)^H$ respectively denote the conjugate, the transpose and the conjugate transpose of the matrix or vector. $\|\cdot\|_2$ refers to the Euclidian norm of the vector. The modulus of the complex number $w$ is denoted by $|w|$. $j = \sqrt{-1}$ denotes the imaginary unit. $\mathbf{I}_m$ represents the $m \times m$ identity matrix, and $\mathbb{C}^{m \times n}$ and $\mathbb{R}^{m \times n}$ represent the sets of $m \times n$ complex matrix and real matrix, respectively. We use $E[\cdot]$ to represent the expected value operator. $\mathrm{diag}(\cdot)$ indicates returning the elements of the main diagonal of the matrix as a vector. $\odot$ and $\otimes$ represent the Hadamard product and Kronecker product, respectively. $\mathbf{A} \succcurlyeq \mathbf{B}$ means $\mathbf{A} - \mathbf{B}$ is a positive semidefinite, i.e., the eigenvalues of $\mathbf{A} - \mathbf{B}$ are nonnegative. Finally, $\mathrm{tr}(\mathbf{A})$ represents the trace of a square matrix $\mathbf{A}$.

## 2. Signal Model of FDA-MIMO Radar

Consider a colocated FDA-MIMO radar, as shown in Figure 1, in which the transmitter and receiver are uniform linear arrays (ULA) with $N$ and $M$ elements, respectively. The element spacing of the transmitter and the receiver is $d = \lambda_0/2$, $\lambda_0 = c/f_0$ is the wavelength, $c$ represents the speed of light, and $f_0$ is the reference carrier frequency. The frequency offset across the transmitter elements is $\Delta f$.

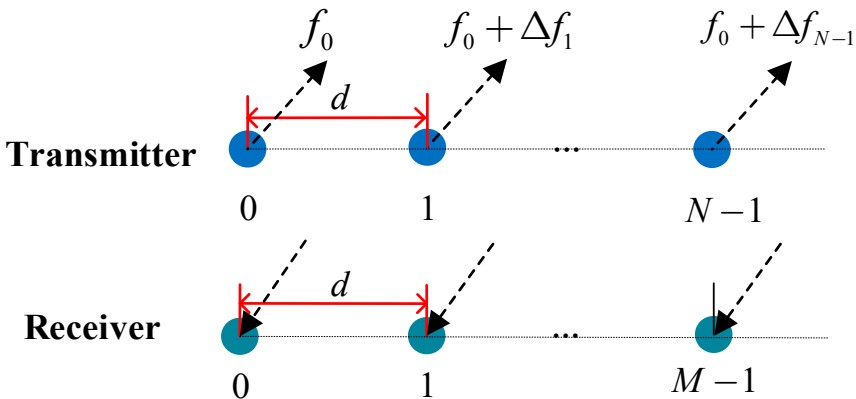

**Figure 1.** The FDA-MIMO array structure.

Thus, the carrier frequency of the $n$-th transmitter element can be written as

$$f_n = f_0 + n\Delta f, \; n = 0, 1 \ldots, N-1. \tag{1}$$

The transmit signal of the $n$-th element can be expressed as

$$s_n(t) = \sqrt{\frac{E}{N}} \, \psi_n(t) e^{j2\pi(f_0 + n\Delta f)t}, \; 0 \le t \le T. \tag{2}$$

where $E$ denotes the transmitted energy, $T$ is the radar pulse duration and $\psi_n(t)$ is the baseband envelope of the $n$-th transmit element, i.e., $\int_0^T \psi_{n_1}^*(t)\psi_{n_1}(t)dt = 1$, which is satisfied with the orthogonality condition

$$\int_0^T \psi_{n_1}^*(t)\psi_{n_2}(t-\tau)e^{j2\pi\Delta f(n_2-n_1)t}dt = 0, n_1 \neq n_2, \forall\tau, \tag{3}$$

where, $\tau$ is the delay time, $n_1, n_2 = 0, 1 \ldots, N-1$.

Assuming that a far-field target is located at $(r_0, \theta_0)$, $\tau_{n,m} = \frac{2r_0}{c} - \frac{nd \sin\theta_0}{c} - \frac{md \sin\theta_0}{c}$ is the echo delay time from the $n$-th transmit element to the $m$-th receive element, and we set $\tau_0 = \frac{2r_0}{c}$. Then, the received signal of the $m$-th element can be expressed as

$$\begin{aligned}
x_m(t) &= \xi \sum_{n=0}^{N-1} \psi_n(t - \tau_{n,m}) e^{j2\pi(f_0 + n\Delta f)(t - \tau_{n,m})} \\
&= \xi \sum_{n=0}^{N-1} \psi_n(t - \tau_{n,m}) e^{j2\pi(f_0 + n\Delta f)(t - \tau_0 + \frac{nd\sin\theta_0}{c} + \frac{md\sin\theta_0}{c})} \\
&= \xi \sum_{n=0}^{N-1} \psi_n(t - \tau_{n,m}) e^{j2\pi f_0(t - \tau_0)} e^{j2\pi f_0 \frac{nd\sin\theta_0}{c}} \times e^{j2\pi f_0 \frac{md\sin\theta_0}{c}} e^{j2\pi n\Delta f(t - \tau_0)} e^{j2\pi n\Delta f(\frac{nd\sin\theta_0}{c} + \frac{md\sin\theta_0}{c})} \\
&\approx \xi e^{j2\pi f_0(t - \tau_0)} e^{j2\pi f_0 \frac{md\sin\theta_0}{c}} \sum_{n=0}^{N-1} \psi_n(t - \tau_0) e^{j2\pi f_0 \frac{nd\sin\theta_0}{c}} e^{j2\pi n\Delta f(t - \tau_0)}
\end{aligned} \tag{4}$$

where the approximation is due to $\frac{n\Delta f}{c}(nd\sin\theta_0 + md\sin\theta_0) \ll 1$ [35,36]. $\xi$ represents the target echo complex coefficient (determined by the target reflection coefficient, space propagation coefficient, amplitude, phase of the transmit signal, etc.). It should be pointed out that this paper mainly studies the beamforming method, which is independent of the target doppler. The doppler factor is ignored for the sake of simplicity.

The signal processing chain of the receiver is shown in Figure 2. The received signal $x_m(t)$ by the $m$-th receiving element is first mixed with $e^{-j2\pi f_0 t}$ in the analog device, and the output signal after mixed is

$$\widetilde{x}_m(t) = \xi e^{-j2\pi f_0 \tau_0} e^{j2\pi f_0 \frac{md\sin\theta_0}{c}} \sum_{n=0}^{N-1} \psi_n(t - \tau_0) e^{j2\pi f_0 \frac{nd\sin\theta_0}{c}} e^{j2\pi n\Delta f(t - \tau_0)}. \tag{5}$$

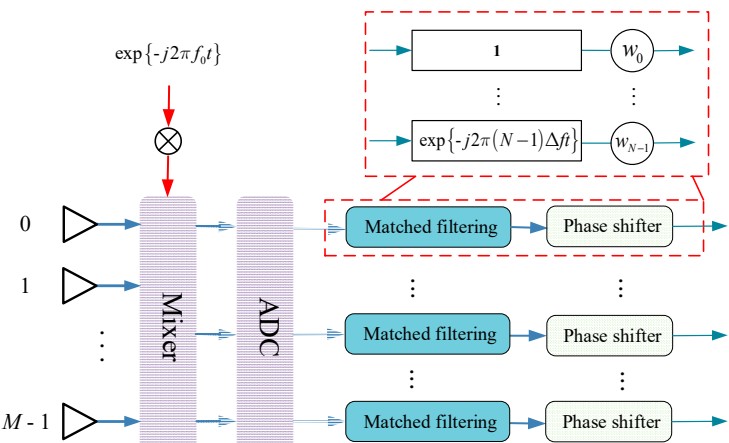

**Figure 2.** The signal processing chain at the receiver array.

After the mixed output signal $\widetilde{x}_m(t)$ is matched with $\phi_n(t)$ in the matched filter, $\phi_n(t)$ is defined as

$$\phi_n(t) = \psi_n(t) e^{-j2\pi n\Delta f t}. \tag{6}$$

According to Equation (3), after the matched filtering, the $n$-th output signal of the $m$-th receive element can be expressed as

$$\hat{x}_{n,m}(t) = \zeta(t) e^{-j2\pi f_0 \tau_0} e^{j2\pi f_0 \frac{md\sin\theta_0}{c}} e^{j2\pi f_0 \frac{nd\sin\theta_0}{c}} e^{-j2\pi n\Delta f \tau_0}, \tag{7}$$

where $\zeta$ represents the complex-valued coefficient after matched filtering.

After matched filtering is performed on the $M$ elements, the FDA-MIMO radar received target echo signal at time $t$ can be expressed as

$$\begin{aligned}
\mathbf{x}_s(t) &= [\hat{x}_{0,0}(t), \cdots, \hat{x}_{0,N-1}(t), \hat{x}_{1,1}(t), \cdots, \hat{x}_{M-1,N-1}(t)]^\mathrm{T} \\
&= \gamma_0 \boldsymbol{\Gamma}_0 \odot [\mathbf{b}_r(\theta_0) \otimes \mathbf{a}_t(r_0, \theta_0)]
\end{aligned} \tag{8}$$

where $\gamma_0 = e^{-j2\pi f_0 \tau_0}$ and $\boldsymbol{\Gamma}_0 \in \mathbb{C}^{NM \times 1}$ is the complex vector after matched filtering. $\mathbf{a}_t(r_0, \theta_0) \in \mathbb{C}^{N \times 1}$ and $\mathbf{b}_r(\theta_0) \in \mathbb{C}^{M \times 1}$ indicate the transmit and receive steering vectors, respectively, which have the forms of

$$\mathbf{a}_t(r_0, \theta_0) = \left[1, e^{j2\pi f_{t0}}, \cdots, e^{j2\pi(N-1)f_{t0}}\right]^{\mathrm{T}}, \tag{9}$$

$$\mathbf{b}_r(\theta_0) = \left[1, e^{j2\pi f_{r0}}, \cdots, e^{j2\pi(M-1)f_{r0}}\right]^{\mathrm{T}}, \tag{10}$$

where $f_{t0} = \frac{d \sin \theta_0}{\lambda} - \frac{2r_0 \Delta f}{c}$ and $f_{r0} = \frac{d \sin \theta_0}{\lambda}$ denote the transmit and receive spatial frequencies, respectively.

Suppose the swarm interference is located in area $\boldsymbol{\Theta}_J$, which consists of numerous mutually independent deceptive interferences (as shown the Figure 3). Similarly, for the interference located at $(r_j, \theta_j)((r_j, \theta_j) \subseteq \boldsymbol{\Theta}_J, j = 1, \ldots, J)$, the received signal after matched filtering can be expressed as

$$\mathbf{x}_j(t) = \gamma_j \boldsymbol{\Gamma}_j \odot \left[\mathbf{b}_r(\theta_j) \otimes \mathbf{a}_t(r_j, \theta_j)\right], \tag{11}$$

where $\gamma_j = e^{-j2\pi f_0 \tau_j}$ and $\boldsymbol{\Gamma}_j \in \mathbb{C}^{NM \times 1}$ denotes the $j$-th complex vector after matched filtering.

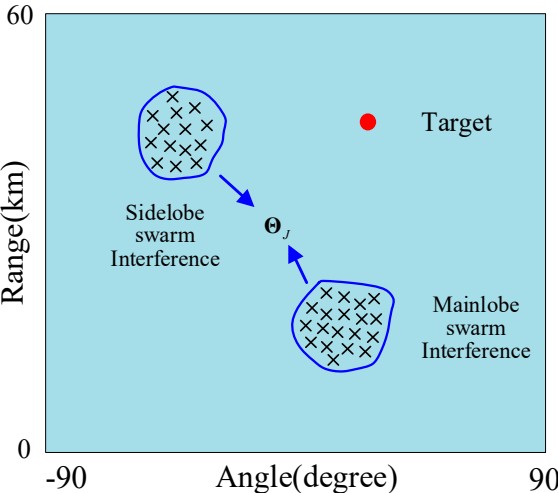

**Figure 3.** Swarm interference model.

In this circumstance, the received signal of the FDA-MIMO radar at time $t$ can be expressed as

$$\mathbf{x}(t) = \mathbf{x}_s(t) + \sum_{j=1}^{J} \mathbf{x}_j(t) + \mathbf{n}(t), \tag{12}$$

where $\mathbf{n}(t) \in \mathbb{C}^{NM \times 1}$ denotes the white Gaussian noise signal with zero mean and covariance matrix $\sigma_n^2 \mathbf{I}$.

## 3. Analysis of Adaptive Beamforming in Interference Suppression

In order to present the data-independent phase-only beamforming method, we briefly recall the adaptive beamforming method in this section.

According to Equation (12), the echo covariance matrix $\mathbf{R}$, the target signal covariance matrix $\mathbf{R}_s$ and the interference plus noise covariance matrix $\mathbf{R}_{j+n}$ can be expressed respectively as

$$\mathbf{R} = \mathrm{E}\left[x(t)x^{\mathrm{H}}(t)\right], \tag{13}$$

$$\mathbf{R}_s = \mathrm{E}\left[\mathbf{x}_s(t)\mathbf{x}_s(t)^{\mathrm{H}}\right] = \sigma_s^2 \mathbf{a}(\theta_0, r_0)\mathbf{a}^{\mathrm{H}}(\theta_0, r_0), \tag{14}$$

$$\mathbf{R}_{j+n} = \mathrm{E}\left[\left(\sum_{j=1}^{J}\mathbf{x}_j(t) + \mathbf{n}(t)\right)\left(\sum_{j=1}^{J}\mathbf{x}_j(t) + \mathbf{n}(t)\right)^{\mathrm{H}}\right] = \sum_{j=1}^{J}\sigma_j^2\mathbf{a}(\theta_j, r_j)\mathbf{a}^{\mathrm{H}}(\theta_j, r_j) + \sigma_n^2\mathbf{I}, \quad (15)$$

where $\sigma_s^2$ and $\sigma_n^2$ represent the energy of the target and noise, respectively, $\sigma_j^2$ indicates the energy of the $j$-th interference, and $\mathbf{a}(\theta_0, r_0) = \mathbf{b}_r(\theta_0) \otimes \mathbf{a}_t(r_0, \theta_0)$ and $\mathbf{a}(\theta_j, r_j) = \mathbf{b}_r(\theta_j) \otimes \mathbf{a}_t(r_j, \theta_j)$ denote the steering vector of the target and the $j$-th interference, respectively. Then, $\mathbf{R}$ can be written as

$$\mathbf{R} = \sigma_s^2\mathbf{a}(\theta_0, r_0)\mathbf{a}^{\mathrm{H}}(\theta_0, r_0) + \sum_{j=1}^{J}\sigma_j^2\mathbf{a}(\theta_j, r_j)\mathbf{a}^{\mathrm{H}}(\theta_j, r_j) + \sigma_n^2\mathbf{I}. \quad (16)$$

For a given complex weight vector $\mathbf{w} = [w_1, \ldots, w_{NM}]^{\mathrm{T}} \in \mathbb{C}^{NM \times 1}$, the output signal of the radar can be expressed as $Y(t) = \mathbf{w}^{\mathrm{H}}\mathbf{x}(t)$. Hence, the energy of the output signal is given by

$$\mathrm{E}\left[|Y(t)|^2\right] = \mathbf{w}^{\mathrm{H}}\mathbf{R}_s\mathbf{w} + \mathbf{w}^{\mathrm{H}}\mathbf{R}_{j+n}\mathbf{w}, \quad (17)$$

where $\mathbf{w}^{\mathrm{H}}\mathbf{R}_s\mathbf{w}$ and $\mathbf{w}^{\mathrm{H}}\mathbf{R}_{j+n}\mathbf{w}$ denote the energy of the output target signal and the interference plus the noise signal after beamforming, respectively. Therefore, the output SINR after beamforming can be defined as

$$\mathrm{SINR}_{\mathrm{out}} = \frac{\mathbf{w}^{\mathrm{H}}\mathbf{R}_s\mathbf{w}}{\mathbf{w}^{\mathrm{H}}\mathbf{R}_{j+n}\mathbf{w}}. \quad (18)$$

In order to maximize the $\mathrm{SINR}_{\mathrm{out}}$, the optimal weight vector $\mathbf{w}_{\mathrm{opt}}$ is given by

$$\mathbf{w}_{\mathrm{opt}} = \mu\mathbf{R}_{j+n}^{-1}\mathbf{a}(\theta_0, r_0), \quad (19)$$

where, $\mu$ is a normalization factor that does not affect the $\mathrm{SINR}_{\mathrm{out}}$. In practical applications, $\mathbf{R}_{j+n}$ cannot be accurately obtained. It is usually replaced by the sample covariance matrix $\hat{\mathbf{R}}$, which is given by

$$\hat{\mathbf{R}} = \frac{1}{L}\sum_{l=1}^{L}\mathbf{x}(l)\mathbf{x}^{\mathrm{H}}(l), \quad (20)$$

where $L$ is the number of sampling snapshots. The sample covariance matrix $\hat{\mathbf{R}}$, however, contains the target signal, which causes the SMI beamformer to be distorted in the target position and leads to a significant decrease in $\mathrm{SINR}_{\mathrm{out}}$. To obtain a better weight vector when including the target signal, the INCMR method was proposed in [11]. Nevertheless, this method still relies on the calculation of the echo covariance matrix. When the number of sampling snapshots is small, it is difficult to obtain the precise INCM information. Furthermore, the phase-only weight vector cannot be obtained, the hardware architecture is complicated and high cost is required when implementing this method.

These discussions motivate us to design a beamforming method in the swarm interference scenario, which needs to satisfy the following conditions: (i) does not depend on INCM acquisition; (ii) achieves interference suppression when only the interference area is known; (iii) maximizes the output SINR; (iv) the weight vector is phase-only.

## 4. Data-Independent Phase-Only Beamforming

In the previous section, we briefly reviewed the conventional adaptive beamforming techniques. To design the optimal weight vector, this problem can be equivalent to

$$\max_{\mathbf{w}} \quad \frac{\mathbf{w}^{\mathrm{H}}\mathbf{R}_s\mathbf{w}}{\mathbf{w}^{\mathrm{H}}\mathbf{R}_{j+n}\mathbf{w}}, \quad (21a)$$

$$\mathrm{s.\,t.}\ F[(\theta_s, r_s)|(\theta_0, r_0)] \geq \rho, \quad (\theta_s, r_s) \subset \mathbf{\Theta}_s, \quad (21b)$$

$$F\big[(\theta_j, r_j)\big|(\theta_0, r_0)\big] \leq \gamma, \quad (\theta_j, r_j) \subset \mathbf{\Theta}_J, \tag{21c}$$

where $\mathbf{\Theta}_s$ denotes the area around the target, $\rho$ indicates the lower bound level constraint to guarantee the mainlobe gain, and $\gamma$ represents the upper bound level constraint to suppress the swarm interference. $F\big[(\theta, r)\big|(\theta_0, r_0)\big]$ represents the normalized array response at $(\theta, r)$, which is defined as

$$F\big[(\theta, r)\big|(\theta_0, r_0)\big] = \frac{\big|\mathbf{w}^H \mathbf{a}(\theta, r)\big|^2}{\big|\mathbf{w}^H \mathbf{a}(\theta_0, r_0)\big|^2}. \tag{22}$$

Therefore, the problem (21) can be considered equivalent to

$$\max_{\mathbf{w}} \quad \frac{\mathbf{w}^H \mathbf{R}_s \mathbf{w}}{\mathbf{w}^H \mathbf{R}_{j+n} \mathbf{w}}, \tag{23a}$$

$$\text{s. t.} \quad \frac{\big|\mathbf{w}^H \mathbf{a}(\theta_s, r_s)\big|^2}{\big|\mathbf{w}^H \mathbf{a}(\theta_0, r_0)\big|^2} \geq \rho, \quad (\theta_s, r_s) \subset \mathbf{\Theta}_s, \tag{23b}$$

$$\frac{\big|\mathbf{w}^H \mathbf{a}(\theta_j, r_j)\big|^2}{\big|\mathbf{w}^H \mathbf{a}(\theta_0, r_0)\big|^2} \leq \gamma, \quad (\theta_j, r_j) \subset \mathbf{\Theta}_J. \tag{23c}$$

*4.1. Constant Modulus Constraint*

In this subsection, we present a data-independent phase-only beamforming method based on constant modulus constraint (CMC). When the modulus of the weight vector is constrained to $h$, the problem (23) can be converted into

$$\max_{\mathbf{w}} \quad \frac{\mathbf{w}^H \mathbf{R}_s \mathbf{w}}{\mathbf{w}^H \mathbf{R}_{j+n} \mathbf{w}}, \tag{24a}$$

$$\text{s. t.} \quad \frac{\big|\mathbf{w}^H \mathbf{a}(\theta_s, r_s)\big|^2}{\big|\mathbf{w}^H \mathbf{a}(\theta_0, r_0)\big|^2} \geq \rho, \quad (\theta_s, r_s) \subset \mathbf{\Theta}_s, \tag{24b}$$

$$\frac{\big|\mathbf{w}^H \mathbf{a}(\theta_j, r_j)\big|^2}{\big|\mathbf{w}^H \mathbf{a}(\theta_0, r_0)\big|^2} \leq \gamma, \quad (\theta_j, r_j) \subset \mathbf{\Theta}_J, \tag{24c}$$

$$|w_i| = h, \quad i = 1, \ldots, NM. \tag{24d}$$

According to the definition of $\mathbf{R}_s$ and $\mathbf{R}_{j+n}$, the $\text{SINR}_{\text{out}}$ can be expressed as

$$
\begin{aligned}
\text{SINR}_{\text{out}} &= \frac{\mathbf{w}^H \mathbf{R}_s \mathbf{w}}{\mathbf{w}^H \mathbf{R}_{j+n} \mathbf{w}} = \frac{\sigma_s^2 \big|\mathbf{w}^H \mathbf{a}(\theta_0, r_0)\big|^2}{\sigma_n^2 \|\mathbf{w}\|_2^2 + \sum\limits_{j=1}^{J} \sigma_j^2 \big|\mathbf{w}^H \mathbf{a}(\theta_j, r_j)\big|^2} \\
&= \frac{1}{\dfrac{\sigma_n^2}{\sigma_s^2} \dfrac{\|\mathbf{w}\|_2^2}{\big|\mathbf{w}^H \mathbf{a}(\theta_0, r_0)\big|^2} + \sum\limits_{j=1}^{J} \dfrac{\sigma_j^2}{\sigma_s^2} \dfrac{\big|\mathbf{w}^H \mathbf{a}(\theta_j, r_j)\big|^2}{\big|\mathbf{w}^H \mathbf{a}(\theta_0, r_0)\big|^2}}
\end{aligned}
\tag{25}
$$

To maximize the $\text{SINR}_{\text{out}}$, we need to set a very small $\gamma$ to suppress the interference. Then, the problem of maximizing $\text{SINR}_{\text{out}}$ can be equivalent to

$$\max_{\mathbf{w}} \ \text{SINR}_{\text{out}} \Leftrightarrow \max_{\mathbf{w}} \big|\mathbf{w}^H \mathbf{a}(\theta_0, r_0)\big|^2. \tag{26}$$

**Proof.** See Appendix A. $\square$

For (26), we expect $\max_{\mathbf{w}} \left| \mathbf{w}^{\mathrm{H}} \mathbf{a}(\theta_0, r_0) \right|^2 = 1$ to guarantee the mainlobe gain. Therefore, the maximization problem (26) is converted into a minimization problem $\min_{\mathbf{w}} \quad 1 - \left| \mathbf{w}^{\mathrm{H}} \mathbf{a}(\theta_0, r_0) \right|^2$. The problem (24) can be considered equivalent to

$$\min_{\mathbf{w}} \quad 1 - \left| \mathbf{w}^{\mathrm{H}} \mathbf{a}(\theta_0, r_0) \right|^2, \tag{27a}$$

$$\text{s. t.} \left| \mathbf{w}^{\mathrm{H}} \mathbf{a}(\theta_s, r_s) \right|^2 \geq \rho, \quad (\theta_s, r_s) \subset \mathbf{\Theta}_{\mathrm{s}}, \tag{27b}$$

$$\left| \mathbf{w}^{\mathrm{H}} \mathbf{a}(\theta_j, r_j) \right|^2 \leq \gamma, \quad (\theta_j, r_j) \subset \mathbf{\Theta}_{\mathrm{J}}, \tag{27c}$$

$$|w_i| = h, \quad i = 1, \dots, NM. \tag{27d}$$

We define $\mathbf{A}_{(\theta, r)} = \mathbf{a}(\theta, r) \mathbf{a}^{\mathrm{H}}(\theta, r) \in \mathbb{C}^{NM \times NM}$ and $\mathbf{W} = \mathbf{w} \mathbf{w}^{\mathrm{H}} \in \mathbb{C}^{NM \times NM}$, and using the semidefinite relaxation (SDR) technique [37], we obtain

$$\begin{aligned} \left| \mathbf{w}^{\mathrm{H}} \mathbf{a}(\theta, r) \right|^2 &= \mathbf{w}^{\mathrm{H}} \mathbf{a}(\theta, r) \mathbf{a}^{\mathrm{H}}(\theta, r) \mathbf{w} \\ &= \mathrm{tr}\left( \mathbf{a}(\theta, r) \mathbf{a}^{\mathrm{H}}(\theta, r) \mathbf{w} \mathbf{w}^{\mathrm{H}} \right) \\ &= \mathrm{tr}\left( \mathbf{A}_{(\theta, r)} \mathbf{W} \right) \end{aligned} \tag{28}$$

In such a way, (27b) and (27c) can be converted to $\mathrm{tr}\left( \mathbf{A}_{(\theta_s, r_s)} \mathbf{W} \right) \geq \rho$ and $\mathrm{tr}\left( \mathbf{A}_{(\theta_j, r_j)} \mathbf{W} \right) \leq \gamma$, respectively. Furthermore, Equation (27d) is constrained by $\mathbf{W} = \mathbf{w} \mathbf{w}^{\mathrm{H}}$ and $|w_i| = p$, and it can be written as $[\mathrm{diag}(\mathbf{W})]_i = h^2$. Thus, problem (27) can be expressed as

$$\min_{\mathbf{w}} \quad 1 - \mathrm{tr}\left( \mathbf{A}_{(\theta_0, r_0)} \mathbf{W} \right), \tag{29a}$$

$$\text{s. t.} \ \mathrm{tr}\left( \mathbf{A}_{(\theta_s, r_s)} \mathbf{W} \right) \geq \rho, \quad (\theta_s, r_s) \subset \mathbf{\Theta}_{\mathrm{s}}, \tag{29b}$$

$$\mathrm{tr}\left( \mathbf{A}_{(\theta_j, r_j)} \mathbf{W} \right) \leq \gamma, \quad (\theta_j, r_j) \subset \mathbf{\Theta}_{\mathrm{J}}, \tag{29c}$$

$$[\mathrm{diag}(\mathbf{W})]_i = h^2, \ i = 1, \dots, NM, \tag{29d}$$

$$\mathrm{tr}(\mathbf{W}) \geq \alpha, \tag{29e}$$

$$\mathbf{W} \succeq 0. \tag{29f}$$

where $\alpha$ is a small positive value to prevent the solution from being zero. The optimal $\mathbf{W}^{\star}$ can be obtained by using the CVX toolbox to solve problem (29). Then, let the eigendecomposition of $\mathbf{W}^{\star}$ be

$$\mathbf{W}^{\star} = \sum_{i=1}^{NM} \lambda_i \mathbf{q}_i \mathbf{q}_i^{\mathrm{T}}. \tag{30}$$

where $\lambda_1 \geq \lambda_2 \geq \cdots \geq \lambda_{NM} > 0$ denotes the eigenvalue after eigendecomposition, and $\mathbf{q}_i \in \mathbb{C}^{NM \times 1}$ is the eigenvector corresponding to $\lambda_i$. Then, the optimal weight vector $\mathbf{w}^{\star}$ is given by [38]

$$\mathbf{w}^{\star} = \lambda_1 \mathbf{q}_1. \tag{31}$$

Therefore, the main steps of the proposed CMC algorithm are summarized in Algorithm 1. The computational complexity of the CMC algorithm is mainly derived from the calculation of the convex optimization, and the computational complexity of solving the convex optimization of the $NM \times 1$ dimensional vector is $O(N^3 M^3)$. Thus, the computational complexity of the CMC algorithm is $O(N^3 M^3)$.

---

**Algorithm 1:** Proposed CMC Algorithm.

---

1:    **Input:** $\boldsymbol{\Theta}_s$, $\boldsymbol{\Theta}_J$, $\rho$, $\gamma$ and $h$.
2:    Use the convex optimization toolbox to solve problem (29) and obtain $\mathbf{W}^\star$.
3:    Using (30) and (31) yields the optimal weight vector $\mathbf{w}^\star$.
4:    **Output:** the optimal weight vector $\mathbf{w}^\star$.

---

### 4.2. Constant Modulus Decomposition

In the previous subsection, we introduced the CMC algorithm. In this subsection, from a different perspective, we decompose the weight vector into constant modulus complex number and use dual phase receive array to synthesize the phase-only beampattern.

Based on (A8) in Appendix A, we constrain $\left|\mathbf{w}^H\mathbf{a}(\theta_0, r_0)\right|^2 = 1$ to guarantee the mainlobe gain, and the problem $\max\limits_{\mathbf{w}} \ \mathrm{SINR}_{\mathrm{out}}$ can be equivalent to $\min\limits_{\mathbf{w}}\|\mathbf{w}\|_2^2$. Thus, the problem (23) can be written as

$$\min_{\mathbf{w}} \quad \|\mathbf{w}\|_2^2, \tag{32a}$$

$$\text{s. t. } \left|\mathbf{w}^H\mathbf{a}(\theta_0, r_0)\right|^2 = 1, \tag{32b}$$

$$\left|\mathbf{w}^H\mathbf{a}(\theta_s, r_s)\right|^2 \geq \rho, \quad (\theta_s, r_s) \subset \boldsymbol{\Theta}_s, \tag{32c}$$

$$\left|\mathbf{w}^H\mathbf{a}(\theta_s, r_s)\right|^2 \leq \gamma, \quad (\theta_j, r_j) \subset \boldsymbol{\Theta}_J. \tag{32d}$$

Similarly, using the SDR technique, (32) can be transformed into

$$\min_{\mathbf{w}} \quad \mathrm{tr}(\mathbf{W}), \tag{33a}$$

$$\text{s. t. } \mathrm{tr}\left(\mathbf{A}_{(\theta_0, r_0)}\mathbf{W}\right) = 1, \tag{33b}$$

$$\mathrm{tr}\left(\mathbf{A}_{(\theta_s, r_s)}\mathbf{W}\right) \geq \rho, \quad (\theta_s, r_s) \subset \boldsymbol{\Theta}_s, \tag{33c}$$

$$\mathrm{tr}\left(\mathbf{A}_{(\theta_j, r_j)}\mathbf{W}\right) \leq \gamma, \quad (\theta_j, r_j) \subset \boldsymbol{\Theta}_j, \tag{33d}$$

$$\mathrm{tr}(\mathbf{W}) \geq \alpha, \tag{33e}$$

$$\mathbf{W} \succeq 0. \tag{33f}$$

Likewise, the problem (33) can be solved using the CVX toolbox to acquire the optimal $\mathbf{W}^\star$. The optimal weight vector $\mathbf{w}^\star$ can be obtained through Equations (30) and (31). Notice that the optimal weight vector $\mathbf{w}^\star$ is not phase-only, and we intend to convert it into phase-only beamforming by designing a dual-phase shifter receiver. Before that, we present the following lemma.

**Lemma 1.** *For any complex number $p = ge^{j\omega}\,(0 \leq g \leq 2b)$, $\omega$ and $g$ denote the phase and modulus of the complex number $p$, respectively; $p$ can be decomposed into*

$$p = be^{j\alpha_1} + be^{j\alpha_2}, \tag{34}$$

*where $\alpha_1 = \omega - \cos^{-1}\frac{g}{2b}$ and $\alpha_2 = \omega + \cos^{-1}\frac{g}{2b}$.*

**Proof** . See Appendix B. □

The above analysis shows that any complex number can be decomposed into two complexes with equal modulus and unequal phase. This means that we can use a dual-

phase shifter at the receiver to synthesize the initial weight vector equivalently. We use Lemma 1 (set $b = w_m^{\star}$) to decompose $w_i^{\star}$, which can be expressed as

$$w_i^{\star} = b e^{j\alpha_{p1}^{\star}} + b e^{j\alpha_{p2}^{\star}}, \tag{35}$$

where $w_m^{\star} = \max\left\{ \left| w_1^{\star} \right|, \left| w_2^{\star} \right|, \cdots, \left| w_{NM}^{\star} \right| \right\}$,

$$\alpha_{p1}^{\star} = \omega_i^{\star} - \cos^{-1} \frac{\left| w_i^{\star} \right|}{2b}, \tag{36}$$

$$\alpha_{p2}^{\star} = \omega_i^{\star} - \cos^{-1} \frac{\left| w_i^{\star} \right|}{2b}, \tag{37}$$

$\omega_i^{\star}$ denotes the phase of the complex number $w_i^{\star}$. After all elements of $\mathbf{w}^{\star}$ have been decomposed, we obtain

$$\mathbf{w}^{\star} = \mathbf{w}_{p1}^{\star} + \mathbf{w}_{p2}^{\star}, \tag{38}$$

where $\mathbf{w}_{p1}^{\star}$ and $\mathbf{w}_{p2}^{\star}$ are phase-only weight vectors. Therefore, we can design the signal processing flow shown in Figure 4 at the receiver.

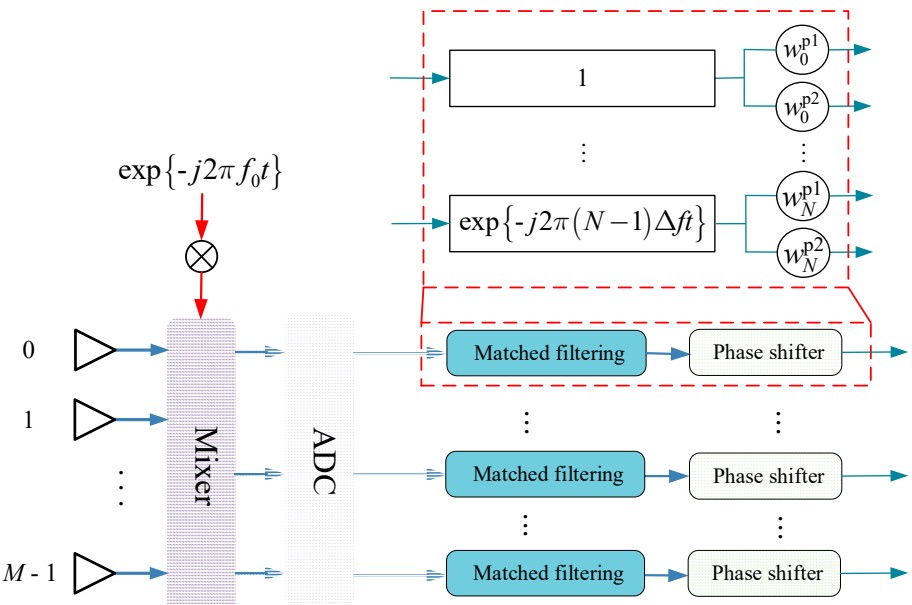

**Figure 4.** Receiver processing flow of the CMD method.

It should be pointed out that after applying Equation (38), other beamforming methods such as CNBC, SNBC and PBN-BF can also be applied to this array structure. These beamforming methods can be transformed into phase-only beamforming. We implement this in the simulation. The procedure of the constant modulus decomposition (CMD) algorithm is given in Algorithm 2. Similarly, the computational complexity of the CMD algorithm is mainly derived from the calculation of the convex optimization. In addition, the dimensionality of the solved weight vector is $NM \times 1$. Thus, the computational complexity of the CMD algorithm is $O(N^3 M^3)$.

---

**Algorithm 2:** Proposed CMD Algorithm.

---

1:   **Input:** $\mathbf{\Theta}_s$, $\mathbf{\Theta}_J$, $\rho$ and $\gamma$.
2:   Using the convex optimization toolbox to solve problem (33) yields $\mathbf{W}^{\star}$.
3:   Using (30) and (31) yields the optimal weight vector $\mathbf{w}^{\star}$.
4:   Use (35) and (38) to perform constant mode decomposition of $\mathbf{w}^{\star}$.
5:   **Output:** the phase-only weight vectors $\mathbf{w}_{\text{p1}}^{\star}$ and $\mathbf{w}_{\text{p2}}^{\star}$.

---

Comparing the CMC and CMD algorithms, we can observe that the CMC algorithm solves the weight vector using the convex optimization with the constant modulus constraint, so that the phase-only weight vector can be acquired directly. It can use the receiver as shown in Figure 2 to synthesize the beampattern. Comparatively, the CMD algorithm achieves the phase-only weight vector by constant modulus decomposition. It requires the dual-phase shifter receiver as shown in Figure 4 to synthesize the beampattern. Furthermore, since the CMC algorithm contains constant-mode constraints, its feasible domain is smaller than the CMD algorithm. Thus, the performance of the CMD algorithm is better than the CMC algorithm, as will be demonstrated in the simulation.

## 5. Simulation Results

In this section, simulation experiments are performed to substantiate the performance of the proposed methods. The main simulation parameters are listed in Table 1. The convex optimization problem is solved by the CVX toolbox in MATLAB [39].

**Table 1.** Simulation parameters.

| Parameters | Symbols | Value | Parameters | Symbols | Value |
|---|---|---|---|---|---|
| Transmit elements | $N$ | 15 | Target range | $r_0$ | 45 km |
| Receive elements | $M$ | 15 | Target angle | $\theta_0$ | 35° |
| Carrier frequency | $f_0$ | 10 GHz | Frequency offset | $\Delta f$ | 1500 Hz |
| Element spacing | $d$ | 0.015 m | Coefficient | $\alpha$ | 0.1 |

### 5.1. Beampattern Comparison for Different Algorithms

In this subsection, we compare the beampattern of different methods to verify the performance of the proposed algorithms. For comparison purposes, the INCMR-linear constraint sector suppressed (LCSS) algorithm [9], SMI-MVDR algorithm [10], SNBC algorithm [29], and CNBC algorithm [29] are simulated. The swarm interference is distributed in an area $\mathbf{\Theta}_J$, $\mathbf{\Theta}_J = \left\{ (\theta_j, r_j) \big| 30° \leq \theta_j \leq 40°, 10 \text{ km} \leq r_j \leq 20 \text{ km} \right\}$. The swarm interference power is 30dB and the target power is 10dB. The desired beampattern level is expected to be lower than $-70$ dB at the swarm interference area.

Figure 5 plots the 2D beampattern of different methods. Figure 6a,b shows cross-sectional views of the 2D beampattern of Figure 5 at $\theta = 35°$ and $r = 45$ km, respectively. It can be seen from Figure 5 that both proposed methods can form the regional deep null in the swarm interference area and effectively suppress the swarm interference. Furthermore, the beam shape of the proposed methods is better than the other methods. It can be seen from Figure 6 that the CMD method has the lowest beampattern level in the interference area and less distortion in the uncontrolled area. The mainlobe width of the two proposed methods is the same as the SNBC method but better than the CNBC method.



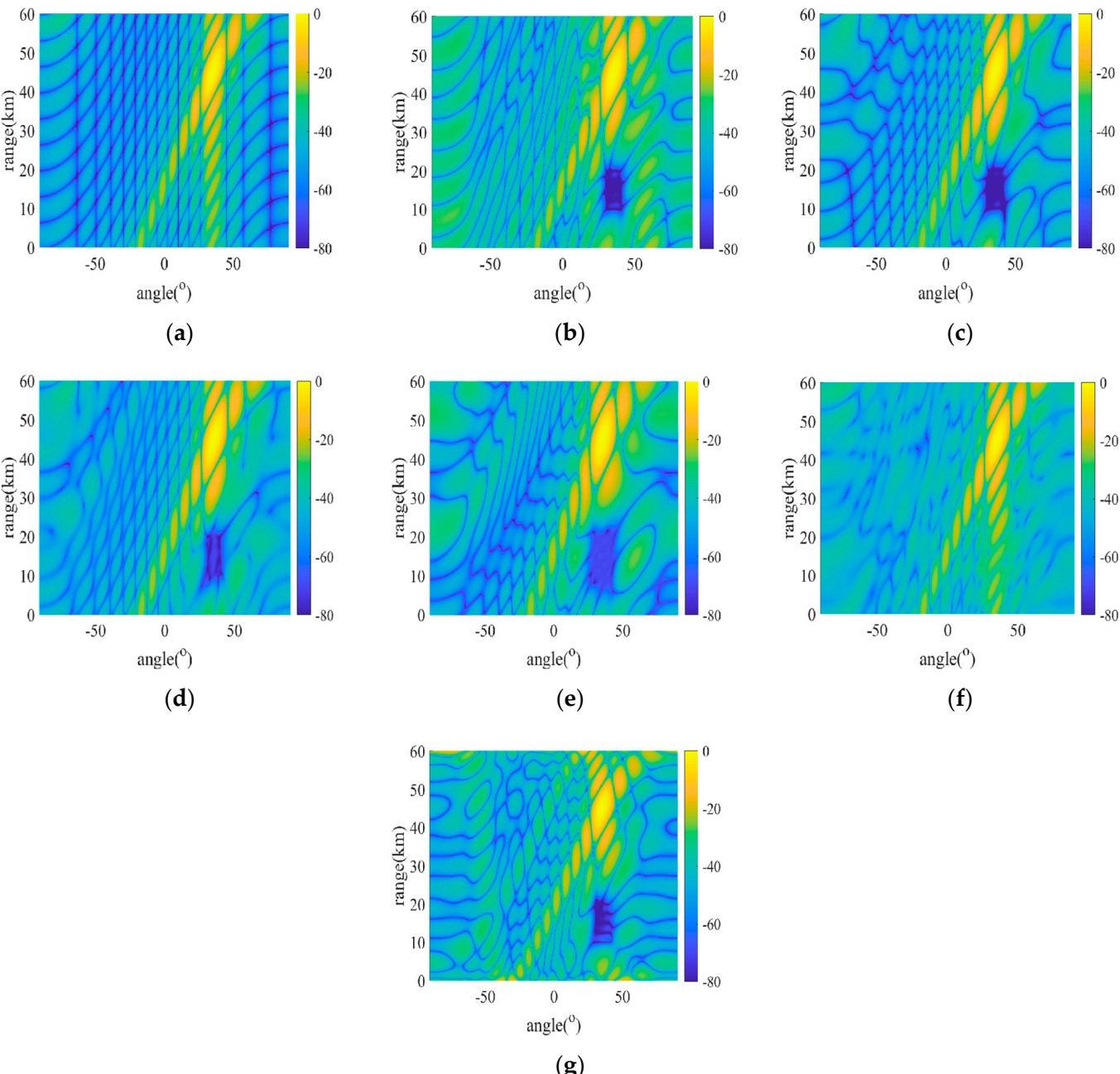

**Figure 5.** Two–dimensional beampattern of different methods when the swarm interference is distributed in an area. (**a**) Initial beampattern, (**b**) CMC, (**c**) CMD, (**d**) SNBC, (**e**) CNBC, (**f**) SMI-MVDR, and (**g**) INCMR-LCSS.

Moreover, we compare the beampattern obtained by different methods when the swarm interference is distributed in two areas, where $\Theta_J = \Theta_J^1 \cup \Theta_J^2$, $\Theta_J^1$ and $\Theta_J^2$ are given by $\Theta_J^1 = \{ (\theta_j, r_j) | 30° \le \theta_j \le 40°, \ 10 \text{ km} \le r_j \le 20 \text{ km} \}$, $\Theta_J^2 = \{ | -5° \le \theta_j \le 5°, \ 10 \text{ km} \le r_j \le 20 \text{ km} \}$. Figure 7 shows the 2D beampattern of different methods. It can be seen that under the two swarm interference areas, two proposed methods can form the regional deep null in the interference area. The sidelobe level of the CMD method is lower than the CMC method.

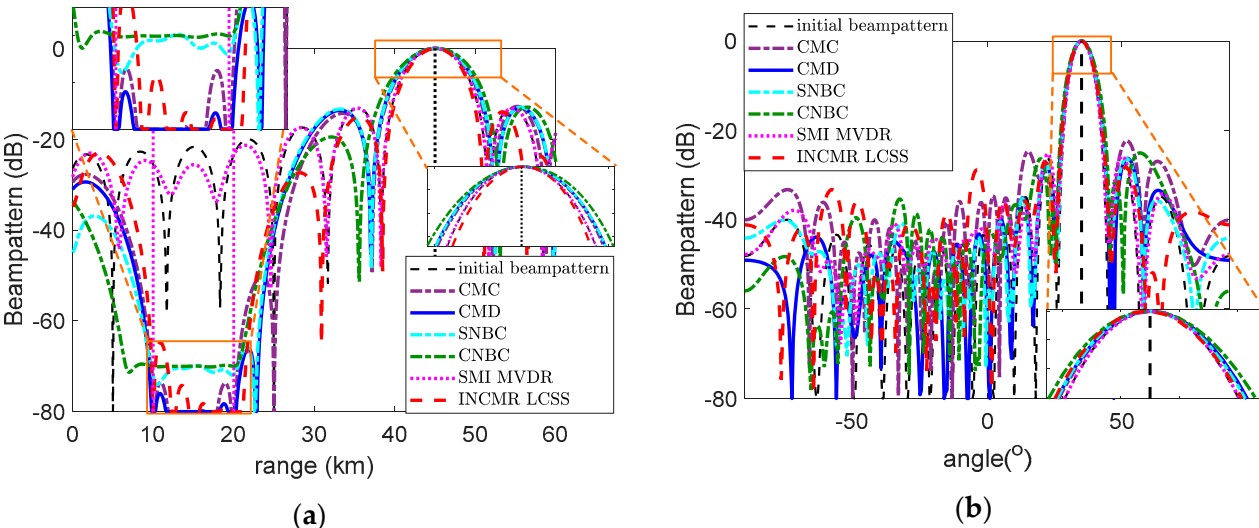

**Figure 6.** The beampattern cross–sectional of different methods at target range and angle. (**a**) The beampattern on $\theta = 35°$. (**b**) The beampattern on $r = 45$ km.

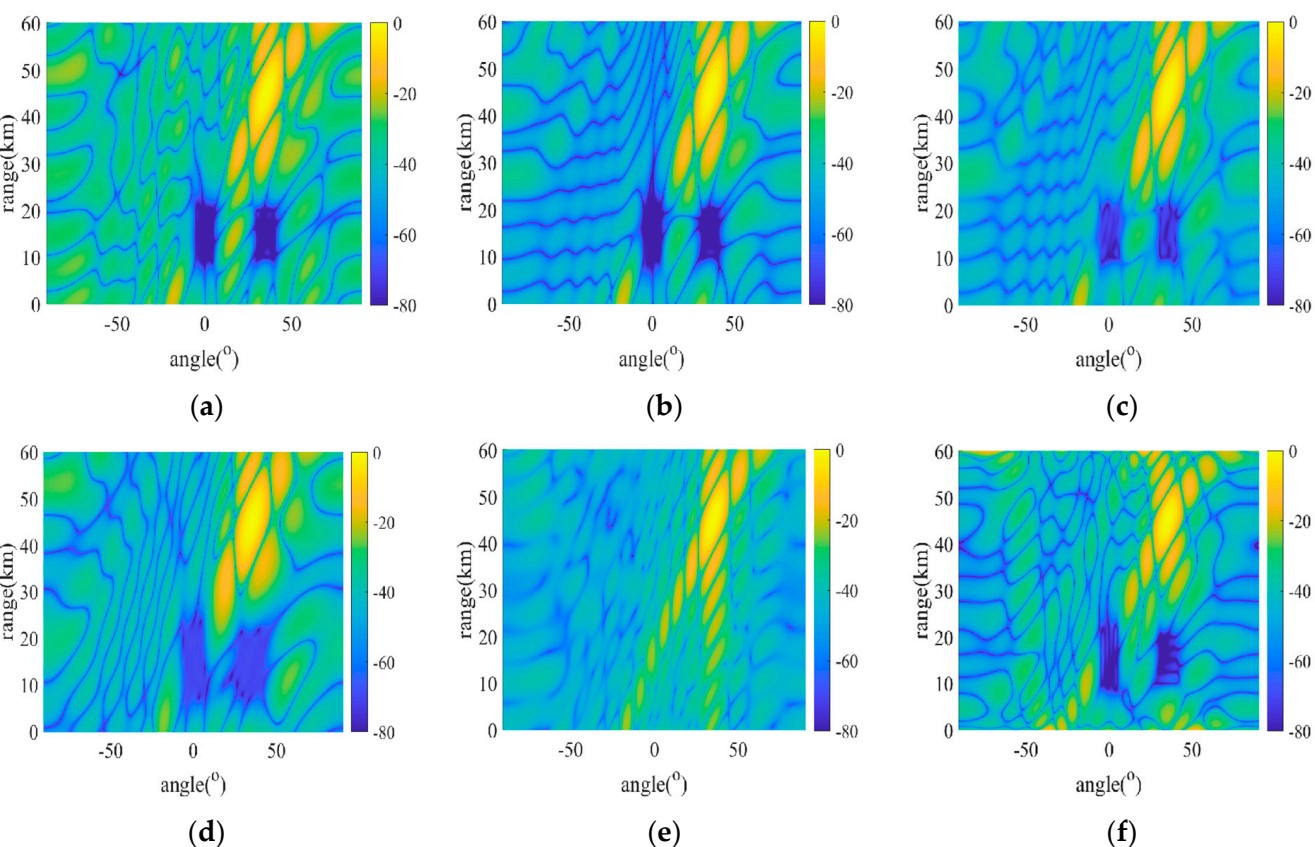

**Figure 7.** Two–dimensional beampattern of different methods when the swarm interference is distributed in two areas. (**a**) CMC, (**b**) CMD, (**c**) SNBC, (**d**) CNBC, (**e**) SMI-MVDR, and (**f**) INCMR–LCSS.

### 5.2. Comparison of the Output SINR

To verify the performance of the proposed method, in this subsection, we compare the output SINR of different methods at different SNR and the number of snapshots. The swarm interference is distributed in an area $\mathbf{\Theta}_J = \left\{ \left( \theta_j, r_j \right) \middle| 30° \leq \theta_j \leq 40°, 10 \text{ km} \leq r_j \leq 20 \text{ km} \right\}$.

Figure 8a shows the output SINR of different beamformers in the SNR from −10 dB to 30 dB when the number of snapshots is 200. Furthermore, under the SNR is 20 dB. Figure 8b shows the output SINR of different beamformers in the number of snapshots from 200 to 1000. In Figure 8a, it can be found that the output SINR increases with the input SNR. From Figure 8b, it can be seen that the proposed methods can obtain excellent output SINR when the number of snapshots is small. In addition, since the proposed methods are data-independent, it can be found that the output SINR does not change with the number of snapshots and maintains good performance. Moreover, as the objective function is to maximize the output SINR, it can be observed that the two proposed methods outperform other methods. The performance of the CMD method is better than the CMC method because its feasible domain is bigger than the CMC algorithm.

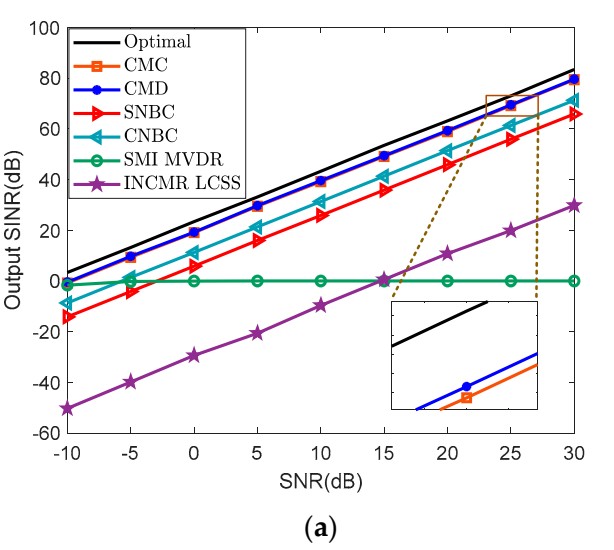
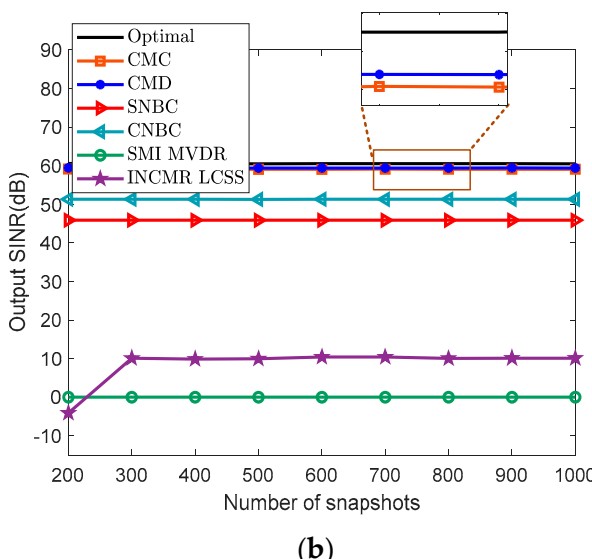

**(a)**                                    **(b)**

**Figure 8.** The output SINR of different methods. (**a**) The output SINR versus SNR. (**b**) The output SINR versus the number of snapshots.

### 5.3. Beampattern on the Different Quantization Bits

In this subsection, we compare the beampattern performance of phase-only methods on different quantization bits. Figures 9 and 10 show the beampattern of the CMC and CMD method when the phase shifter has different quantization bits, respectively. In addition, Figures 11 and 12 plot the beampattern of CNBC and SNBC after constant modulus decomposition on the dual-phase shifter receiver with different quantization bits, respectively. Furthermore, it can be found from Figures 9 and 10 that the synthesis beampattern is significantly different from the original beampattern when the phase shifters have different quantization bits. Figures 11 and 12 demonstrate that the CMD method can be applied to the conventional beamforming method and converted into phase-only beamforming. It can be observed that the beampattern can reach the performance of the original beampattern as the number of quantization bits of the phase shifter increases. When the quantization bit is 3, the beampattern of the interference area is different from the original beampattern, but the mainlobe beampattern remains suitable. With eleven quantization bits, the quantized beampattern is approximate to the original beampattern.

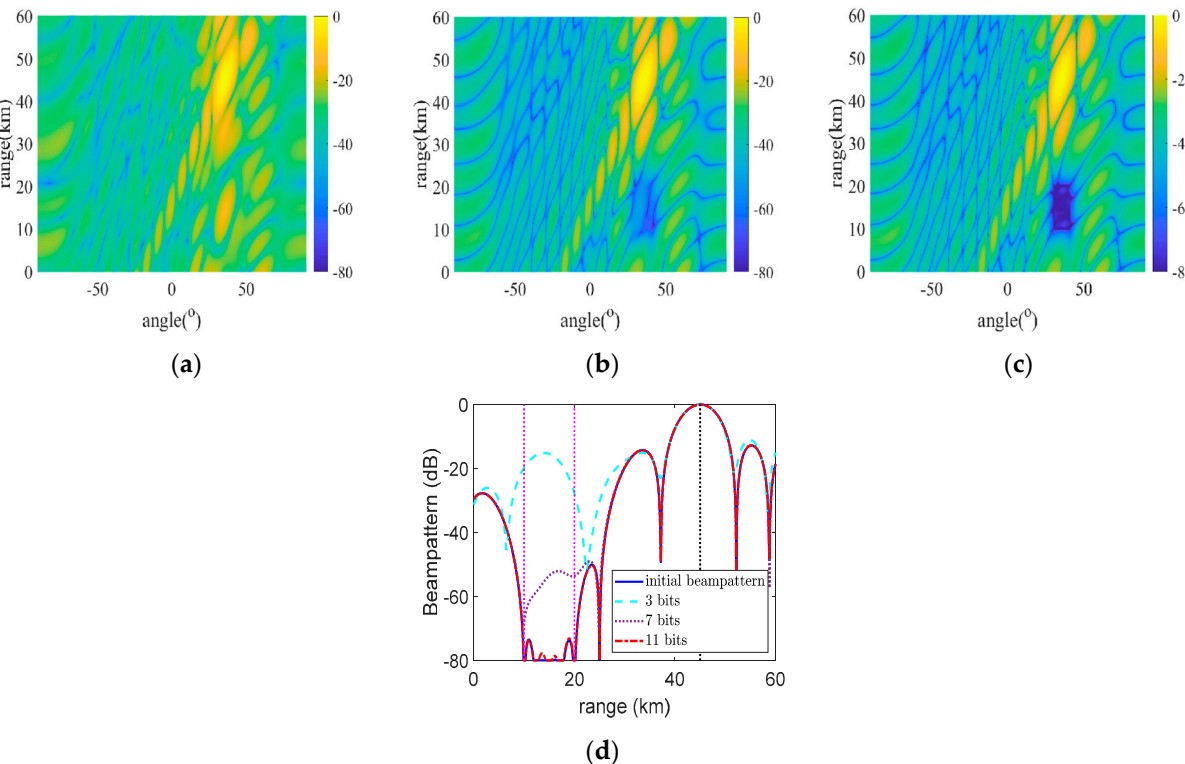

**Figure 9.** The CMC beampattern on different quantization bits: (**a**) 3 bits, (**b**) 7 bits, (**c**) 11 bits, (**d**) cross–section of different quantization at target angle.

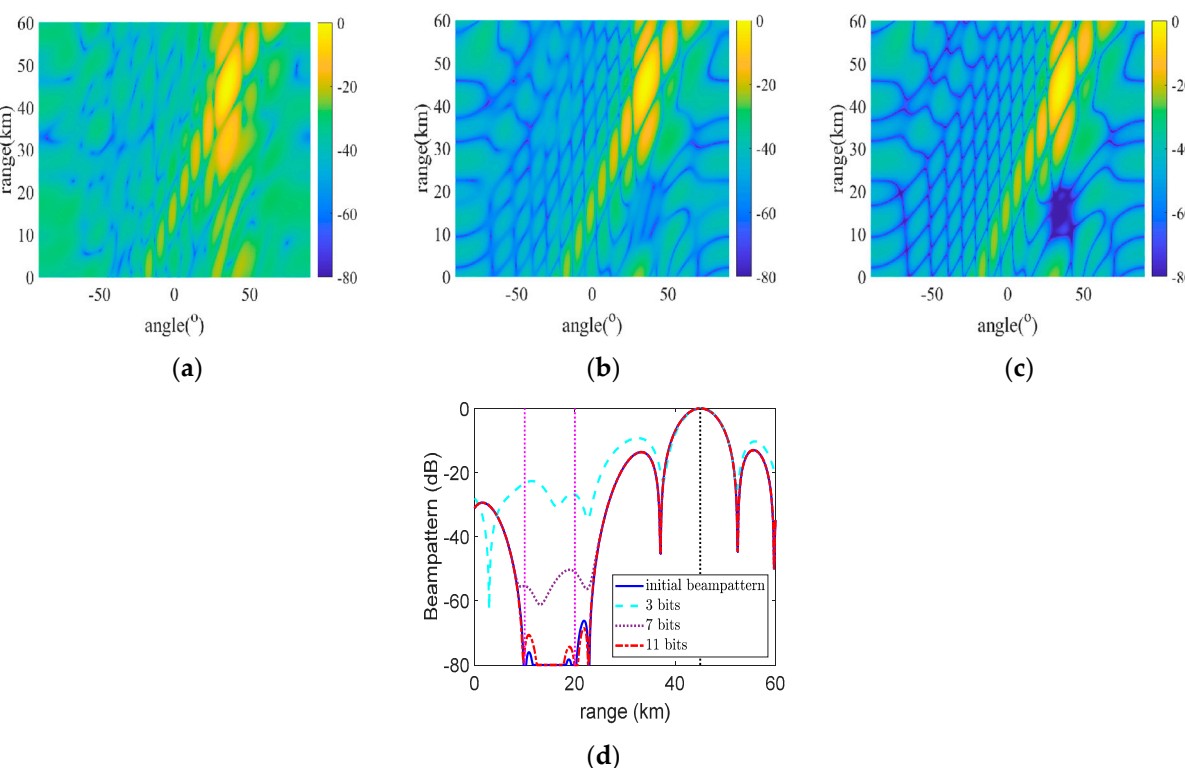

**Figure 10.** The CMD beampattern on different quantization bits: (**a**) 3 bits, (**b**) 7 bits, (**c**) 11 bits, (**d**) cross–section of different quantization at target angle.

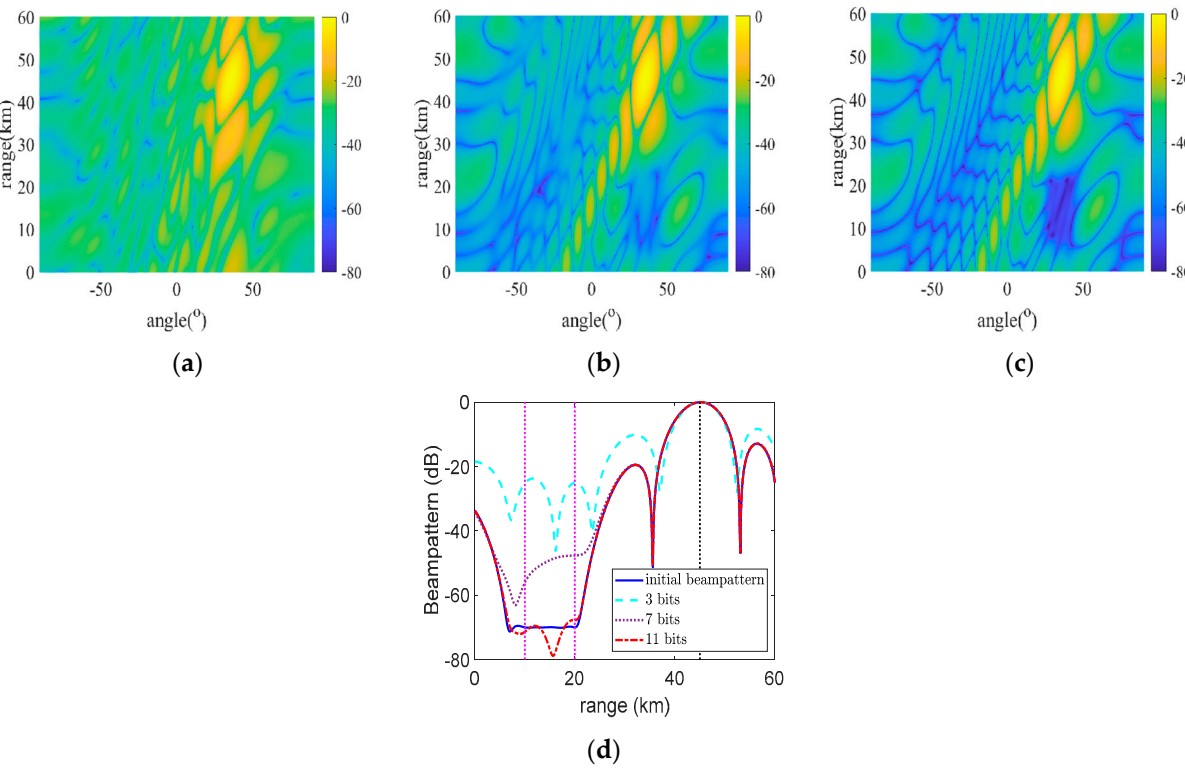

**Figure 11.** The constant modulus decomposition CNBC beampattern on different quantization bits: (**a**) 3 bits, (**b**) 7 bits, (**c**) 11 bits, (**d**) cross–section of different quantization at target angle.

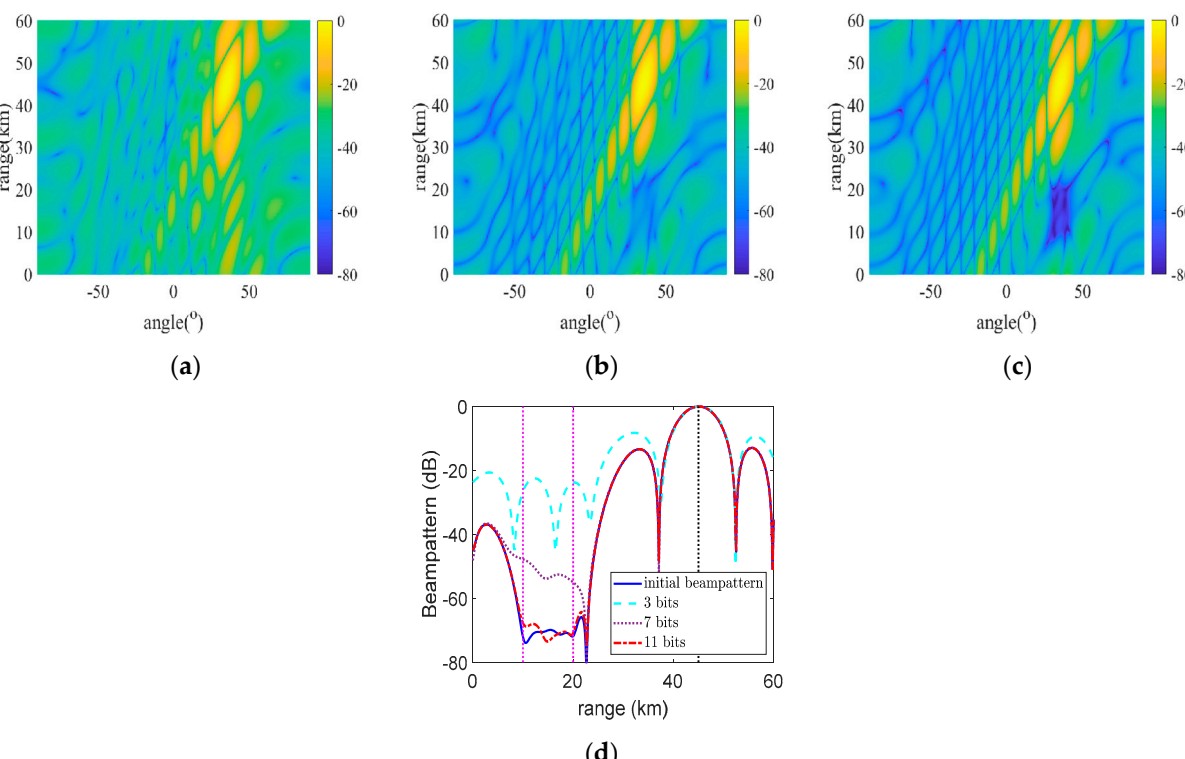

**Figure 12.** The constant modulus decomposition SNBC beampattern on different quantization bits: (**a**) 3 bits, (**b**) 7 bits, (**c**) 11 bits, (**d**) cross–section of different quantization at target angle.

### 5.4. Output SINR on the Different Quantization Bits

In this subsection, we compare the output SINR of phase-only methods on different quantization bits. Figures 13 and 14 plot the output SINR of the CMC, CMD, Phase-only (PO) CNBC and PO SNBC methods on different quantization bits, respectively. It can be seen that as the number of quantization bits of the phase shifter increases, the output SINR can approximate the original performance. When the quantization bit is 9, the quantized output SINR is very different from the original output SINR. The output SINR can approach the original performance when the quantization bits reach 13.

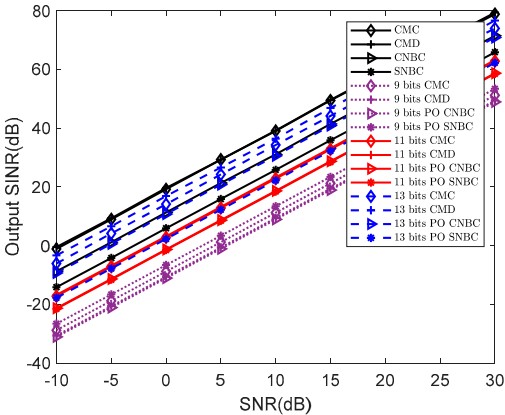

**Figure 13.** The output SINR versus SNR on different quantization bits.

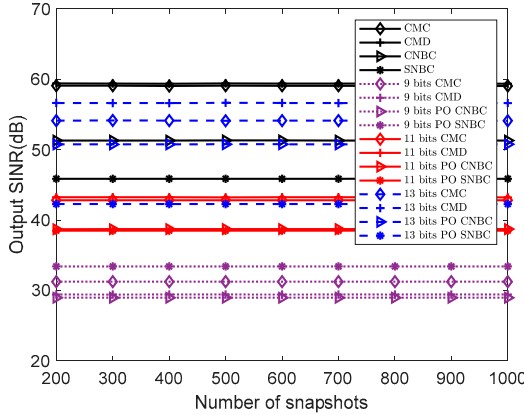

**Figure 14.** The output SINR versus the number of snapshots on different quantization bits.

### 6. Conclusions

In this paper, we proposed two data-independent phase-only beamforming methods for FDA-MIMO radar to suppress the swarm interference. The two proposed algorithms can suppress swarm interference effectively. Firstly, we convert the output SINR into the data-independent form by imposing the constant modulus constraint and interference area level constraint. The phase-only weight vector is solved by maximizing the output SINR as the objective function. Secondly, from a different perspective, we proposed the CMD beamforming method. This method decomposed the weight vector into constant modulus complex number and used the dual-phase shifter receiver to synthesize the phase-only beampattern. It is worth emphasizing that the CMD method can also be applied to other complex weight beamforming methods to achieve phase-only beampattern synthesis. Simulation experiments demonstrate the superiority of the proposed method. As a future work, we shall consider how to reduce the computational complexity of the proposed methods.

**Author Contributions:** G.C.: methodology, software, writing—original draft. C.W.: conceptualization, investigation—review & editing. J.G.: formal analysis, funding acquisition, writing—review & editing. M.T.: methodology, supervision, validation, writing—review & editing. Y.L.: validation, writing—review & editing. All authors have read and agreed to the published version of the manuscript.

**Funding:** This work is supported in part by the National Nature Science Foundation of China under Grant 62201580 and by the Natural Science Foundation of Shaanxi Province under Grants 2021JM-222 and 2023-JC-YB-553.

**Data Availability Statement:** The data used to support the findings of this study are available from the corresponding author upon request.

**Conflicts of Interest:** The authors declare no conflict of interest.

## Appendix A

**Proof** ($\max\ \mathrm{SINR_{out}} \Leftrightarrow \max\ |\mathbf{w}^H\mathbf{a}(\theta_0, r_0)|^2$).

Firstly, the maximization problem (25) is converted into a minimization problem:

$$\min_{\mathbf{w}}\quad \frac{\sigma_n^2}{\sigma_s^2}\frac{\|\mathbf{w}\|_2^2}{|\mathbf{w}^H\mathbf{a}(\theta_0, r_0)|^2} + \sum_{j=1}^{J}\frac{\sigma_j^2}{\sigma_s^2}\frac{|\mathbf{w}^H\mathbf{a}(\theta_j, r_j)|^2}{|\mathbf{w}^H\mathbf{a}(\theta_0, r_0)|^2}. \tag{A1}$$

According to constraint (24c), we have

$$\sum_{j=1}^{J}\frac{\sigma_j^2}{\sigma_s^2}\frac{|\mathbf{w}^H\mathbf{a}(\theta_j, r_j)|^2}{|\mathbf{w}^H\mathbf{a}(\theta_0, r_0)|^2} \le \gamma\sum_{j=1}^{J}\frac{\sigma_j^2}{\sigma_s^2}. \tag{A2}$$

Thus, we can obtain

$$\frac{\sigma_n^2}{\sigma_s^2}\frac{\|\mathbf{w}\|_2^2}{|\mathbf{w}^H\mathbf{a}(\theta_0, r_0)|^2} \le \frac{\sigma_n^2}{\sigma_s^2}\frac{\|\mathbf{w}\|_2^2}{|\mathbf{w}^H\mathbf{a}(\theta_0, r_0)|^2} + \sum_{j=1}^{J}\frac{\sigma_j^2}{\sigma_s^2}\frac{|\mathbf{w}^H\mathbf{a}(\theta_j, r_j)|^2}{|\mathbf{w}^H\mathbf{a}(\theta_0, r_0)|^2} \le \frac{\sigma_n^2}{\sigma_s^2}\frac{\|\mathbf{w}\|_2^2}{|\mathbf{w}^H\mathbf{a}(\theta_0, r_0)|^2} + \gamma\sum_{j=1}^{J}\frac{\sigma_j^2}{\sigma_s^2}. \tag{A3}$$

Let us set $g(\mathbf{w}) = \frac{\sigma_n^2}{\sigma_s^2}\frac{\|\mathbf{w}\|_2^2}{|\mathbf{w}^H\mathbf{a}(\theta_0, r_0)|^2} + \sum_{j=1}^{J}\frac{\sigma_j^2}{\sigma_s^2}\frac{|\mathbf{w}^H\mathbf{a}(\theta_j, r_j)|^2}{|\mathbf{w}^H\mathbf{a}(\theta_0, r_0)|^2}$, $f(\mathbf{w}) = \frac{\sigma_n^2}{\sigma_s^2}\frac{\|\mathbf{w}\|_2^2}{|\mathbf{w}^H\mathbf{a}(\theta_0, r_0)|^2}$. When $g(\mathbf{w})$ and $f(\mathbf{w})$ are respectively taken as objective functions, they have the same feasible domain due to the same constraints. Suppose $\mathbf{w}^*$ is the optimal solution to $f(\mathbf{w})$, then we have the inequation

$$g(\mathbf{w}^*) \le f(\mathbf{w}^*) + \gamma\sum_{j=1}^{J}\frac{\sigma_j^2}{\sigma_s^2}. \tag{A4}$$

As $f(\mathbf{w}) \le g(\mathbf{w})$, we have

$$-g(\mathbf{w}) \le -f(\mathbf{w}). \tag{A5}$$

Subsequently, we can obtain

$$g(\mathbf{w}^*) - g(\mathbf{w}) \le f(\mathbf{w}^*) - f(\mathbf{w}) + \gamma\sum_{j=1}^{J}\frac{\sigma_j^2}{\sigma_s^2}. \tag{A6}$$

Owing to $f(\mathbf{w}^*) - f(\mathbf{w}) \le 0$, Equation (A6) is rewritten as

$$g(\mathbf{w}^*) - g(\mathbf{w}) \le \gamma\sum_{j=1}^{J}\frac{\sigma_j^2}{\sigma_s^2}. \tag{A7}$$

Therefore, $\mathbf{w}^*$ is the optimal solution that satisfies the error $\gamma \sum_{j=1}^{J} \frac{\sigma_j^2}{\sigma_s^2}$ in the feasible region of $g(\mathbf{w})$. Therefore, $f(\mathbf{w})$ can be used as the objective function when $\gamma$ is very small. In beamforming, we usually set a very small $\gamma$ to suppress the interference. Hence, the objective function (A1) can be equivalent to

$$\min_{\mathbf{w}} \ \frac{\sigma_n^2}{\sigma_s^2} \frac{\|\mathbf{w}\|_2^2}{|\mathbf{w}^H \mathbf{a}(\theta_0, r_0)|^2} + \sum_{j=1}^{J} \frac{\sigma_j^2}{\sigma_s^2} \frac{|\mathbf{w}^H \mathbf{a}(\theta_j, r_j)|^2}{|\mathbf{w}^H \mathbf{a}(\theta_0, r_0)|^2} \Rightarrow \min_{\mathbf{w}} \ \frac{\sigma_n^2}{\sigma_s^2} \frac{\|\mathbf{w}\|_2^2}{|\mathbf{w}^H \mathbf{a}(\theta_0, r_0)|^2}. \tag{A8}$$

where $\frac{\sigma_s^2}{\sigma_n^2}$ denotes the input SNR, which is a constant. Furthermore, $\mathbf{w}$ is constrained by (24d), $\|\mathbf{w}\|_2^2$ is also a constant. Hence, we can obtain

$$\max \ \mathrm{SINR}_{\mathrm{out}} \Leftrightarrow \max \ \left| \mathbf{w}^H \mathbf{a}(\theta_0, r_0) \right|^2. \tag{A9}$$

□

## Appendix B

**Proof of Lemma 1.** For different values of $g$, we can divide it into the following three cases to prove.

(1) When $g = 0$, there is $\alpha_1 - \alpha_2 = \pi$, and the above formula is obviously established.
(2) When $g = 2b$, there is $\alpha_1 = \omega$, $\alpha_2 = \omega$, and then $be^{j\alpha_1} + be^{j\alpha_2} = ge^{j\omega}$.
(3) Consider the common case that $0 < g < 2b$, we plot $p = ge^{j\omega}$ in the complex plane, as shown in Figure A1, where $\overrightarrow{OP}$ denotes the complex $p$ and $\left| \overrightarrow{OP} \right| = g$. After that, find the midpoint $Q$ of the $\overrightarrow{OP}$ such that $\left| \overrightarrow{QP} \right| = \left| \overrightarrow{OQ} \right| = \frac{g}{2}$. We make a vertical line of $\overrightarrow{OP}$ through point $Q$ and intersect the circle of radius $b$ at points $A$ and $B$, i.e., $OP \perp AB$. Therefore, we obviously find $\overrightarrow{OP} = \overrightarrow{OA} + \overrightarrow{OB}$.

According to the geometric relationship in Figure A1, the angle $\delta$ between $\overrightarrow{OA}$ and $\overrightarrow{OQ}$ can be given by

$$\delta = \cos^{-1} \frac{\left| \overrightarrow{OQ} \right|}{\left| \overrightarrow{OA} \right|} = \cos^{-1} \frac{g}{2b}. \tag{A10}$$

Therefore, the phase angle of $\overrightarrow{OA}$ is $\alpha_1 = \omega - \cos^{-1} \frac{g}{2b}$, and the phase angle of $\overrightarrow{OB}$ is $\alpha_2 = \omega + \cos^{-1} \frac{g}{2b}$. Using the fact that $\overrightarrow{OP} = \overrightarrow{OA} + \overrightarrow{OB}$, we obtain

$$p = be^{j(\omega - \cos^{-1} \frac{g}{2b})} + be^{j(\omega + \cos^{-1} \frac{g}{2b})}, \tag{A11}$$

where $b$ can take any value within $b \geq g/2$. In practical implementation, it is more suitable to set $b = g/2$.

Combining the above three cases, Lemma 1 is proved. □

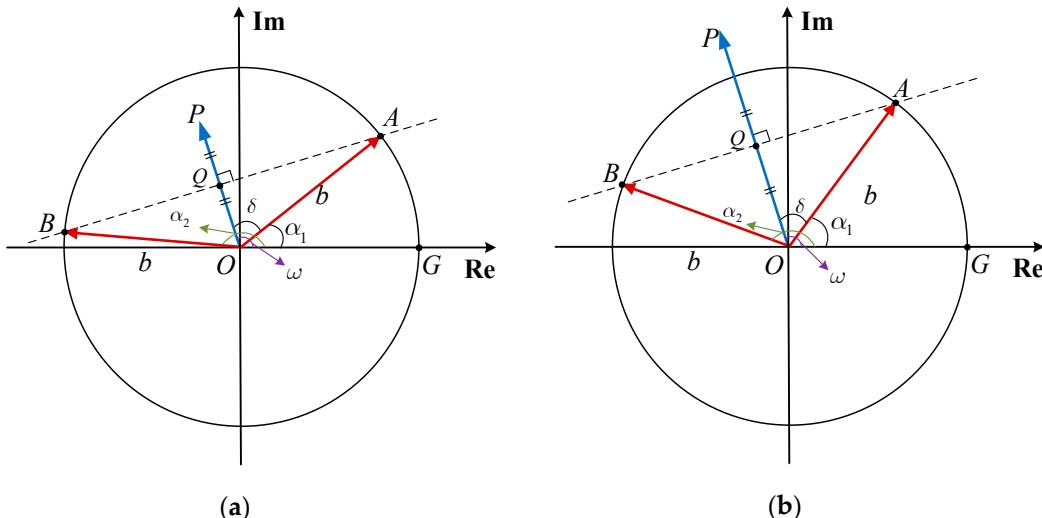

**Figure A1.** The diagram of complex number decomposition. (**a**) $0 < g < b$ ; (**b**) $b \leq g < 2b$.

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
