# Peer review of "Data-Independent Phase-Only Beamforming of FDA-MIMO Radar for Swarm Interference Suppression"

_remotesensing, doi:10.3390/rs15041159_

Round 1
Reviewer 1 Report
The research topic is interesting. However, I have the following concerns:
1. It is unclear why we need to suppress swarm jamming. Is there any different from the normal mainlobe jamming?
2. It is unclear why we need to perform phase-only beamforming in the Rx end. Where is the ADC located at in the receiver? Is the matched filtering conducted in the analog domain?
3. The literature survey on phase-only beamforming is not sufficient.
4. Some introduction on FDA-MIMO radar can be added, such as
[1] Clutter Suppression for Airborne FDA-MIMO Radar using Multi-Waveform Adaptive Processing and Auxiliary Channel STAP. Signal Processing, 2019, 154(1): 280-293.
[2] Enhanced Three-Dimensional Joint Domain Localized STAP for Airborne FDA-MIMO Radar under Dense False-Target Jamming Scenario. IEEE Sensors Journal, 2018, 18(10): 4154-4166.
5. The contributions are unclearly conveyed in this paper.
6. Fig.5: The two phase shifters should be connected in tandem structure instead of parallel structure?
7, In the simulation part, the author assumes the Tx waveforms are orthogonal. However, perfect orthogonal waveform is hard to generate in practice. So, what the performance would be if the non-orthogonal CDMA waveform is used as Tx waveform?
8. The English could be polished.
9. Another problem is that the data-independent beamforming is sensitive to various array errors. How the performance of the proposed method will be if there exists array errors?
Author Response
Thank you for your review of our manuscript, and the response to your question is in the attachment.

Reviewer 2 Report
Interesting paper. It would be nice to see experimental validation
Author Response
Thank you for your approval of our manuscript.
Reviewer 3 Report
We suggest that authors should consider rewriting abstracts, contributing.
a) The author's title and abstract highlight Data-Independent Phase-Only Beamforming, and propose two methods(based on CMC, CMD), but when summarizing contributions, the authors emphasize Data-Independent and Phase-Only Beamforming respectively. Does this mean Data-Independent Phase-Only Beamforming is separated, not jointly ?
b) The author proposes beamforming algorithms based on CMD and CMC respectively, but from the simulation results, the performance of CMD is better than that of CMC, we suggest that the author make a brief statement on the difference between these two algorithms.
The form of the formula and its number should be unified.
The symbol definitions for Eq. 8 should be given, and also for Eq. 11.
The quality of the all figures needs significant improvement with a good presentation, but the Fig 10-11 subtitles are confusing.
Author Response

(The authors gave the same response as above.)

Reviewer 4 Report
This paper proposes two data-independent phase-only beamforming methods of FDA-MIMO radar for swarm interference suppression, one based on constant modulus constraint (CMC) and the other based on constant modulus decomposition (CMD). Simulation experiments demonstrate the effectiveness of the proposed method, but the specific comments are as follows.
1. the format of equations and figures in this paper needs to be adjusted, pay attention to the typography.
2. give the flowchart of the algorithm in this paper.
3. explain the computational efficiency of this method and give the computation time of the algorithm.
4. explain why the processing performance of CMD is better than that of CMC in section 5.2.
5. pay attention to the format of the references.
Author Response

(The authors gave the same response as above.)

Reviewer 5 Report
This article is well structured. Upon careful study of the article, I have no complaints about the work.
Author Response

(The authors gave the same response as above.)

Round 2
Reviewer 1 Report
I have no further comment.